# Organic fertilization co-selects genetically linked antibiotic and metal(loid) resistance genes in global soil microbiome

Zi-Teng Liu [1], Rui-Ao Ma[1], Dong Zhu[2], Konstantinos T. Konstantinidis [3], Yong-Guan Zhu [4,5] & Si-Yu Zhang [1]✉

Antibiotic resistance genes (ARGs) and metal(loid) resistance genes (MRGs) coexist in organic fertilized agroecosystems based on their correlations in abundance, yet evidence for the genetic linkage of ARG-MRGs co-selected by organic fertilization remains elusive. Here, an analysis of 511 global agricultural soil metagenomes reveals that organic fertilization correlates with a threefold increase in the number of diverse types of ARG-MRG-carrying contigs (AMCCs) in the microbiome (63 types) compared to non-organic fertilized soils (22 types). Metatranscriptomic data indicates increased expression of AMCCs under higher arsenic stress, with co-regulation of the ARG-MRG pairs. Organic fertilization heightens the coexistence of ARG-MRG in genomic elements through impacting soil properties and ARG and MRG abundances. Accordingly, a comprehensive global map was constructed to delineate the distribution of coexistent ARG-MRGs with virulence factors and mobile genes in metagenome-assembled genomes from agricultural lands. The map unveils a heightened relative abundance and potential pathogenicity risks (range of 4-6) for the spread of coexistent ARG-MRGs in Central North America, Eastern Europe, Western Asia, and Northeast China compared to other regions, which acquire a risk range of 1-3. Our findings highlight that organic fertilization co-selects genetically linked ARGs and MRGs in the global soil microbiome, and underscore the need to mitigate the spread of these co-resistant genes to safeguard public health.

Soil, as one of Earth's largest reservoirs of microbial diversity, is also a hotspot where antibiotic resistance genes (ARGs) can be transferred to a wide range of pathogens and represents a continued threat to human and animal health and sustainability[1,2]. This phenomenon is even more pronounced in agricultural soils due to the selective pressure of intensive anthropogenic inputs associated with fertilization practices. Due to the growing usage of antibiotics in livestock,

a considerable amount of antibiotics, such as tetracyclines, sulfonamides, quinolones, and macrolides, remain in manure, and thus, likely select for the ARGs present in agricultural soils when organic fertilizer is applied[3–5]. In addition to antibiotics, metal(loid), especially copper (Cu), zinc (Zn), arsenic (As), and chromium (Cr), are also used in livestock to maintain animal health and promote growth and are therefore transferred to the agricultural soils through organic

[1]Shanghai Key Lab for Urban Ecological Processes and Eco-Restoration, School of Ecological and Environmental Sciences, East China Normal University, Shanghai, China. [2]Key Laboratory of Urban Environment and Health, Ningbo Observation and Research Station, Institute of Urban Environment, Chinese Academy of Sciences, Xiamen, China. [3]School of Civil & Environmental Engineering and School of Biological Sciences, Georgia Institute of Technology, Atlanta, GA, USA. [4]State Key Laboratory of Urban and Regional Ecology, Research Center for Eco-Environmental Sciences, Chinese Academy of Sciences, Beijing, China. [5]Institute of Urban Environment, Chinese Academy of Sciences, Xiamen, China. ✉e-mail: syzhang@des.ecnu.edu.cn

fertilizer application[6–8]. The long-term application of manure has also been reported to increase the concentrations of As, mercury (Hg), Cr, Cu, Zn, and manganese (Mn) in agricultural soils[6,9,10]. The imported antibiotics and metal(loid) applied during fertilizer application could further lead to an increase in the diversity and frequency of co-occurring ARG-metal(loid) resistance genes (MRGs), along with an elevated potential for horizontal gene transfer (HGT) of these genes to pathogenic or opportunistic pathogenic organisms and the associated risk of pathogenicity[8]. However, these studies were based on correlations between the abundances of ARGs and MRGs and did not provide evidence for the direct, genetic linkage between ARGs and MRGs in the soil microbiome.

In the mining-impacted environment, which is characterized by high levels of metal(loid), the association between the abundances of ARGs and MRGs has been previously observed, indicating that metal(loid) stresses prevalent in the environment could drive the co-occurrence of ARGs and MRGs. A study of the globally distributed mining-impacted environments showed that multidrug resistance genes tend to co-occur with multimetal resistance genes more frequently as confirmed by genetic associations between ARGs and MRGs[11]. Because several bacterial antibiotic and metal(loid) resistance systems share common modes of action, i.e., the same gene confers resistance to multiple types of antibiotics and metal(loid), or because of the co-presence of the same genetic element resistance such as mobile genetic elements (MGEs), metal(loid) stress may exert long-term and widespread co-selective pressure on antibiotic resistance[12–14]. However, in agricultural soils amended with organic fertilizer, which may also introduce both antibiotics and metal(loid)[15], it is not clear whether the previous associations of ARGs and MRGs found represent spurious results or rather that the ARGs and MRGs are genetically linked; that is, they are carried within the same genetic element such as plasmid DNA. Genetic linkages have been previously observed based on bacterial isolate genomes[13,16], including the coexistence of Zn and beta-lactam resistance, bacitracin and polymyxin resistance, and cadmium (Cd) and aminoglycoside resistance, but it is unclear to what extent these linkages can be found within complex microbial communities such as those occupying agricultural soils. To date, with the increasing number of agricultural soil metagenomic datasets obtained from various countries, an extensive analysis of specific genetic associations of ARGs and MRGs in the agricultural soil microbiome and an evaluation of their global potential risk are achievable and can advance the knowledge gaps mentioned above.

Herein, 511 metagenomic datasets of agricultural soils with or without organic fertilization practices were collected from different countries at the global scale. Genetic linkage analyses were carried out to profile the different coexistence types of ARGs and MRGs in agricultural soil microbiomes under different fertilization practices. We also sampled 12 agricultural soils under different levels of metal(loid) treatment for metagenomic and metatranscriptomic analyses. Considering that the As organic compound roxarsone is one of the most commonly used food additives in livestock to treat parasitic diseases and for animal fattening[17,18], as well as the worldwide reported As contamination in agricultural soils resulting from either manure application or geologic origins[19], the stress of As on the transcriptional activities of the ARGs and MRGs found in the same DNA molecule and/or genome was further assessed. Factors including climate, socioeconomic status, and soil properties, were examined as driving factors of the coexistence of ARGs and MRGs in the agricultural soil microbiome. According to the critical affecting factors, the abundances of ARG-MRG-carrying microbes with potential pathogenicity and gene transmission potential were predicted, and a global map of agricultural lands delineating their distribution was generated to unveil potential health risks from spreading ARGs on different continents using machine learning.

## Results

### Overview of microbial communities and resistomes in agricultural soils at the global scale

A total of 511 agricultural soil metagenomes were distributed across 17 countries (Fig. 1a and Supplementary Table 1), including Australia, Canada, China, Finland, Germany, India, Indonesia, Italy, Japan, Mexico, Russia, Slovenia, South Africa, Switzerland, the United Kingdom, the United States, and Vietnam. An average of 64,623,003 clean reads, 360,427 contigs, and 557,535 ORFs were generated per sample after trimming and assembling. These agricultural soil metagenomes were classified into two groups, i.e., organic fertilization applied (OF) or not applied (NOF) based on the information if available. For the remaining metagenomes without fertilization information, an in-house-built random forest (RF) classification model with an F1 score of 0.97, which indicated a robust result of sample classification, was used for grouping the agricultural soil metagenomes (Supplementary Fig. 1). Ultimately, these agricultural soil metagenomes were classified as 227 NOF and 284 OF samples (Supplementary Fig. 2).

The soil microbial communities in the NOF samples exhibited significantly ($p < 0.001$) greater Shannon diversity than those in the OF samples (Fig. 1b). The significantly greater robustness (two-sided $t$-test, $p < 0.001$; Fig. 1b) and other metrics (Supplementary Fig. 3) in the OF network than in the NOF network indicated a heightened complexity in the interaction of microbial communities in agricultural soils with organic fertilizer application. A significantly ($p < 0.001$) increased diversity (richness) of ARGs, risk ARGs, and MRGs was observed in OF soils relative to that in NOF soils (60.3 vs. 44.1, 5.5 vs. 2.3 and 84.7 vs. 69.1, respectively; Fig. 1c and Supplementary Fig. 4), especially in Europe and/or North America (Supplementary Fig. 5). The relative abundance of risk ARGs was significantly positively correlated with MRGs ($R^2 = 0.10$, $p < 0.001$) and MGEs ($R^2 = 0.19$, $p < 0.001$) in OF soils, while this correlation was not observed in NOF soils (Supplementary Fig. 5). Microbes belonging to *Bacteroidetes*, *Firmicutes*, and *Pseudomonadota*, which exhibited significantly increased abundances in OF soils (adjusted $p < 0.05$, |log2FoldChange| > 1) contributed to the increased abundance and diversity of the risk ARGs and MRGs in OF soils (Supplementary Fig. 6).

### Genetic linkage profiles of ARG-MRG-carrying contigs in the NOF and OF agricultural soil microbiomes

The ARG-MRG-carrying contigs (AMCCs) which indicated the coexistence of ARG-MRG in microbial genomes, showed significantly ($p < 0.001$) higher abundances in OF than NOF soils (0.44 vs. 0.11 copies per cell, i.e., the fraction of total cells encoding the AMCCs). Moreover, a higher fraction of AMCC carrying MGEs (4.59% vs. 0.86%), virulence factor genes (VFGs; 2.72% vs. 0.05%), and both (1.14% vs. 0.98%) was observed in OF than NOF soils (Fig. 2a and Supplementary Fig. 7). For AMCCs carrying MGEs, more various MGE types, including integrative and conjugative element (ICE), actinomycete ICE (AICE), insertion sequence (IS), recombinase, transposon, integrative and mobilizable element (IME), and integron were observed in OF soils, while in NOF soils, only three MGE types (ICE, integron and transposase) were found (Fig. 2b). Approximately 36% and 43% of the AMCCs were predicted to be on the chromosome, and 30% and 17% were predicted to be on the plasmid for the NOF and OF soils, respectively (Fig. 2b). A total of 63 and 22 types of AMCCs were observed in OF and NOF soils, respectively, representing an approximately threefold difference (Fig. 2c, d). Multidrug resistance was the predominant ARGs coexisting with MRGs in AMCCs, accounting for 88% of the total AMCC abundance in NOF and 77% in OF soils (Fig. 2c, d). While the coexistence of multidrug and Zn resistance genes was the main (71%) AMCC type identified in the NOF soils, this AMCC type accounted for 45% of the total AMCC abundance in the OF soils (Fig. 2c, d). Multidrug resistance genes coexisting with aluminum (Al), As, gold (Au), Cd, Cr, Cu, iron (Fe), selenium (Se), and multimetal resistance genes

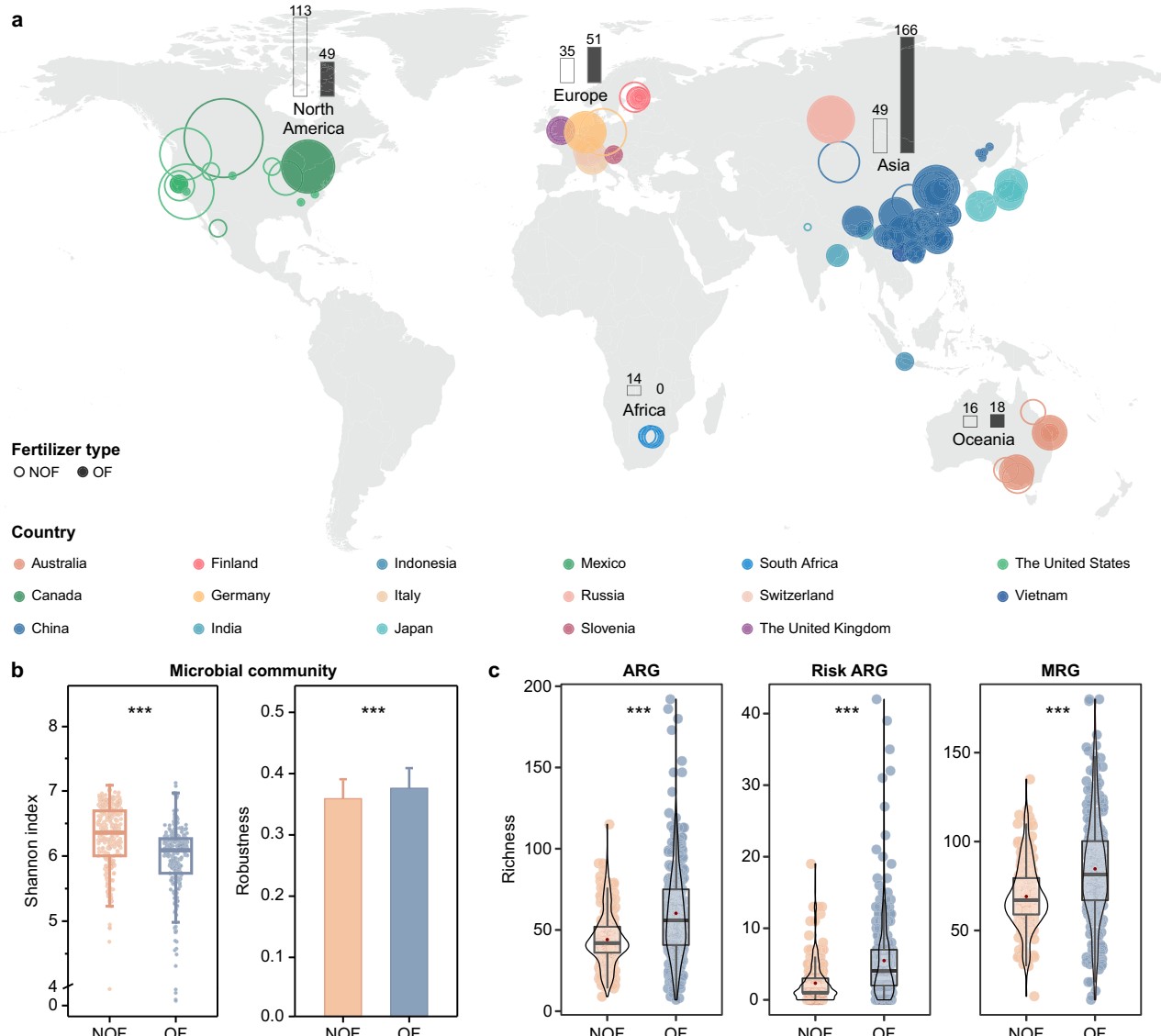

**Fig. 1 | Global distribution of agricultural soil metagenomes and associated diversity of microbial communities and resistomes in organic fertilized (OF) and nonorganic fertilized (NOF) soils. a** Geographic location of globally collected soil metagenomes. Each point indicated one sampling location, with the point size reflecting the number of samples, and the point color indicating the country. Hollow and solid dots represented NOF and OF samples, respectively. Twenty-six samples without latitude and longitude information were not shown. The bar charts indicate the sample numbers of the NOF and OF soils on each continent. **b** Microbial diversity was assessed by Shannon index (two-sided Wilcoxon test, ***$p < 0.001$) in NOF ($n = 227$) and OF ($n = 284$) soils. Robustness was measured as the proportion of taxa remaining after 50% random removal from each network

(Supplementary Fig. 3). Each error bar corresponds to the standard deviation (SD) of 100 repetitions ($n = 100$) of the simulation and data were showed as mean + SD. Significant comparisons (two-sided $t$-test, $t = 4.19$, $df = 198$, ***$p < 0.001$) between NOF and OF were indicated. **c** Diversity (Richness) of antibiotic resistance genes (ARGs), risk ARGs, and metal(loid) resistance genes (MRGs) in the NOF ($n = 227$) and OF ($n = 284$) samples (two-sided Wilcoxon test, ***$p < 0.001$). The boxes indicated the 25th to 75th percentiles (with the median as a horizontal line), and the whiskers represented the maximum and minimum values except for outliers. Each point was a sample in the box plot. In **b**, **c** orange and blue dots represented NOF and OF samples, respectively. Source data were provided as a Source Data file. ARG antibiotic resistance genes, MRG metal(loid) resistance gene.

were identified in OF soils and accounted for a total of 32% of the total AMCCs. Other ARG types including kasugamycin, vancomycin, bacitracin, tetracycline, fosmidomycin, rifamycin, beta-lactam, and quinolone resistance were also observed mostly in OF soils (Fig. 2d).

**ARG-MRG coexistent gene cluster and transcriptional activity evaluation under arsenic stress in agricultural soils**

For the AMCCs obtained from the collected metagenomic datasets, the majority of ARGs (47/71) coexisted with a single specific MRG on the same contig, and the remaining ARGs (24/71) coexisted with multiple MRGs (Fig. 3a). The coexistence of the *mdtA*, *mdtB*, and *mdtC*

genes, which confer resistance to both multidrug and Zn, represented a cross-resistance mechanism[20] that was dominant in the NOF (71%) and in the OF soils but had a relatively lower proportion, i.e., only 45%, of the total coexistent types of ARG-MRGs. The coexistence of variant ARG and MRG subtypes, which are genetically linked on the same MGE[12], were approximately fivefold more abundant in OF than in NOF soils. Additionally, a gene cluster (plasmid) with almost identical genes (with identity and query breadth coverage by amino acids greater than 90%) of As resistant (*acr3*, *arsC2*, *arsH*, and *arsR3*), transposon (*Tn6183*, and *Tn6302*) and sulfonamide resistant (*sul2*) plants adjacent to each other was identified in different OF soils (Fig. 3b), corroborating the coexistence of ARG-MRG in the OF soil microbiome.

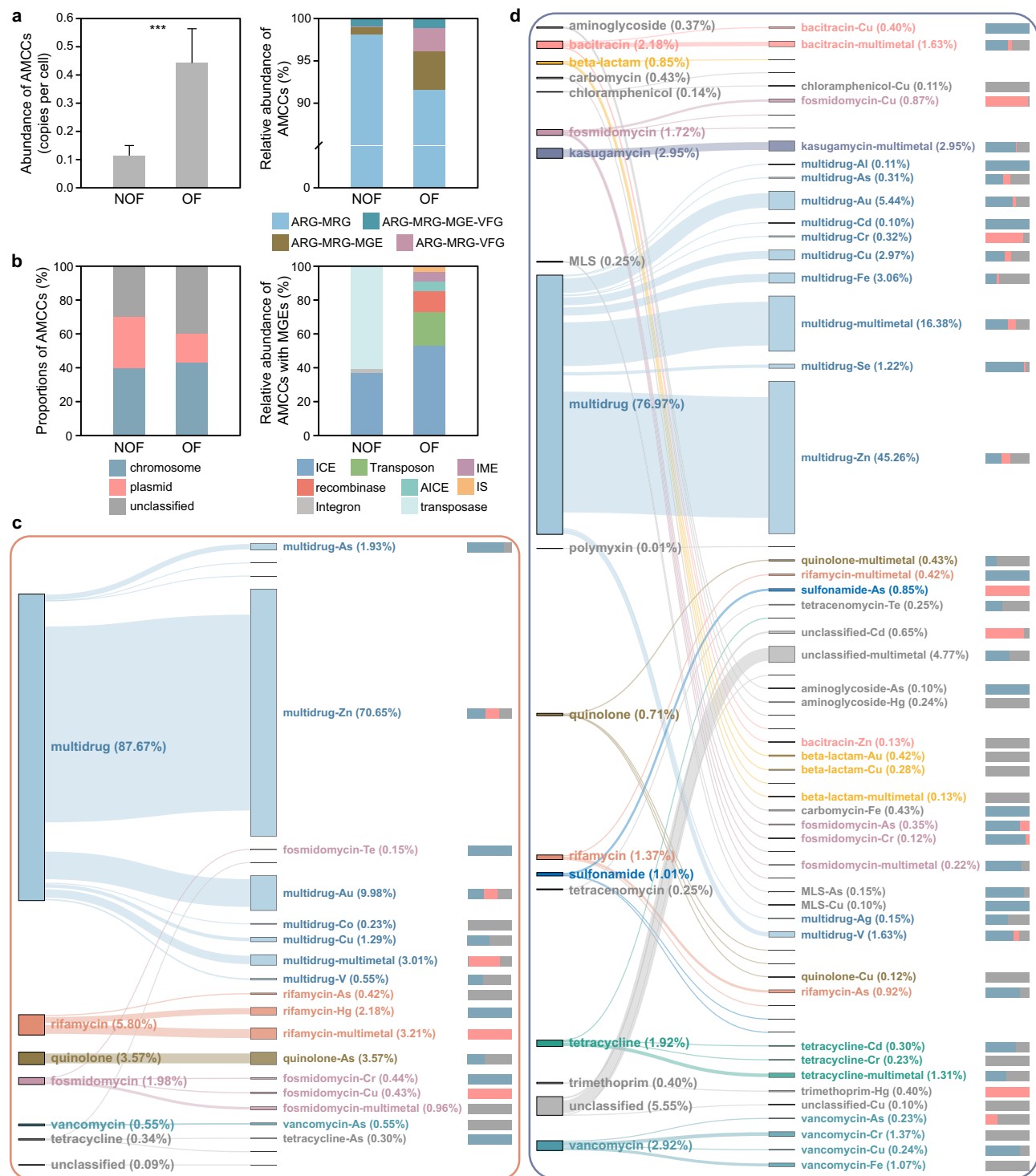

**Fig. 2 | Abundances and types of coexistent ARG-MRGs in agricultural soil microbiomes. a** Abundance and relative proportion of ARG-MRG-carrying contigs (AMCCs) in NOF ($n = 227$) and OF ($n = 284$) soils (two-sided Wilcoxon test, ***$p < 0.001$). Each error bar corresponds to the SD and data were showed as mean + SD. Different colors represented different types of coexistence. **b** The proportions of AMCCs encoded by plasmids or chromosomes in NOF and OF soils. Abundance and relative proportion of AMCCs with mobile genetic elements (MGEs) in the NOF and OF soils (Different colors for different MGE types). **c** Coexistent types of ARGs, MRGs, MGEs, and virulence factor genes (VFGs) on contigs, along with proportions of AMCCs encoded by plasmids or chromosomes in the NOF soils. Blue, pink, and gray represented chromosomes, plasmids, and unclassified sequences, respectively. **d** Coexistent types of ARGs, MRGs, MGEs, and VFGs on contigs, along with proportions of AMCCs encoded by plasmids or chromosomes in the OF soils. In **c**, **d** different colors represented different ARG types, with those with a proportion less than 0.5% and unclassified types set to gray. Source data were provided as a Source Data file. AMCC ARG-MRG-carrying contig, MGE mobile genetic elements, VFG virulence factor gene, ICE integrative and conjugative element, IME integrative and mobilizable element, AICE actinomycete ICE, IS insertion sequence, MLS macrolide-lincosamide-streptogramin.

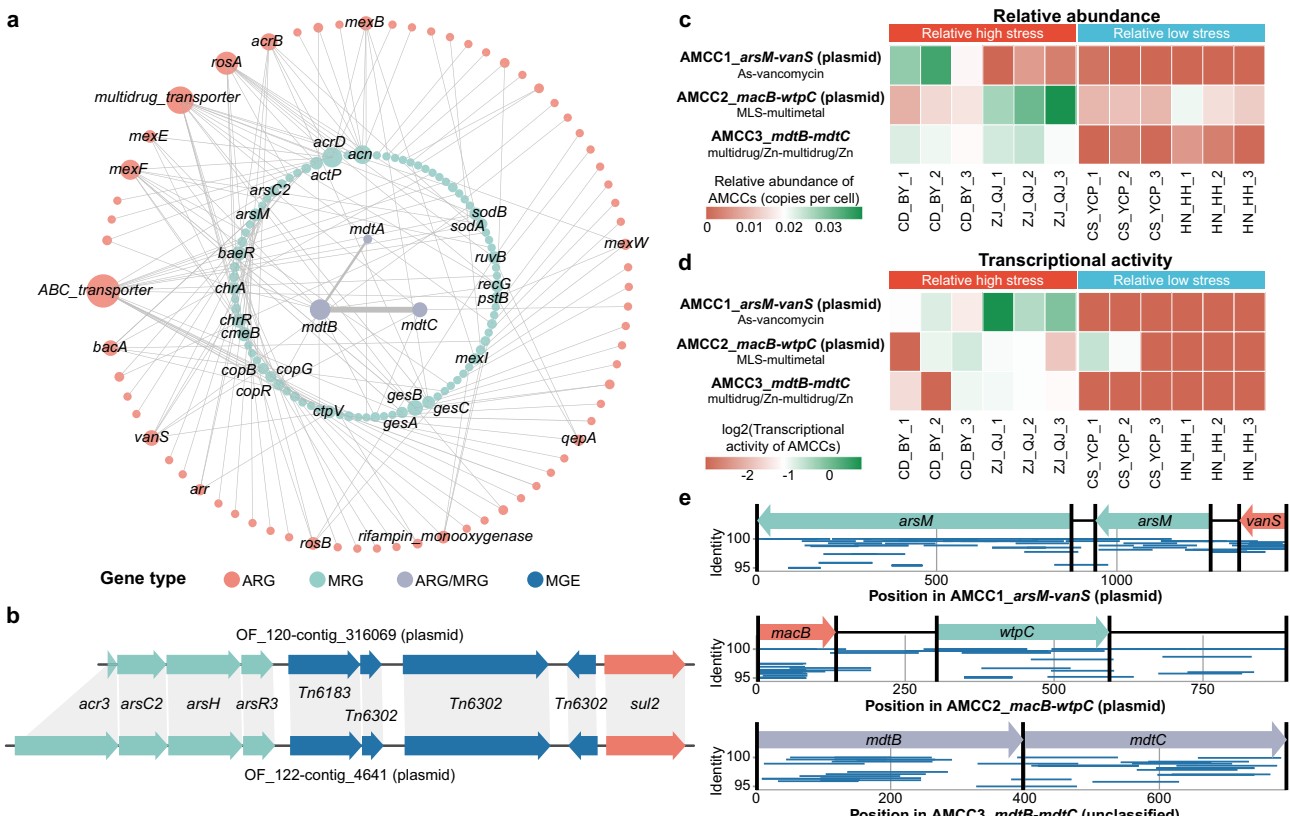

**Fig. 3 | Coexistent ARG-MRG subtypes and associated transcriptional activities in agricultural soil microbiomes. a** Coexistence network between ARG and MRG subtypes, with node size indicating the presence frequency and edge width representing the coexistence frequency of ARG/MRG subtypes on contigs. **b** Representative arrangements of AMCCs on plasmids carrying arsenic resistant (*acr3, arsC2, arsH, arsR3*), transposon (*Tn6183, Tn6302*), and sulfonamide resistant (*sul2*) genes (identity and query breadth coverage by amino acids greater than 90%). Relative abundance (**c**) and transcriptional activity (**d**) of AMCCs in paddy soils with relatively high and low arsenic stress. **e** Recruitment plot of reads from the metatranscriptomic datasets originating from CD_BY_2 (top), ZJ_QJ_1 (middle), and CD_BY_1 (bottom) mapped against the assembled AMCCs. In Fig. 4a, b, e, different colors represented different gene types. Source data were provided as a Source Data file. ARG antibiotic resistance gene, MRG metal(loid) resistance gene, MGE mobile genetic element, AMCC ARG-MRG-carrying contigs.

To further validate the co-selection of antibiotics and As resistance, 12 agricultural soils were sampled from China, and subjected to metagenomic and metatranscriptomic sequencing. According to the National Soil Environmental Quality Standard of China, which suggested soil contamination of As if the total soil As content was higher than 15 mg kg⁻¹, these soil samples were classified into relatively low levels of As stress (average As concentration of 9.32 mg kg⁻¹) and relatively high levels of As stress (average As concentration of 19.49 mg kg⁻¹) groups. The co-resistant AMCCs on plasmids carrying vancomycin-As resistance genes (*vanS-arsM*) and macrolide-lincosamide-streptogramin (MLS)-multimetal resistance genes (*macB-wtpC*), and the ARG-MRG cross-resistant AMCC carrying *mdtB-mdtC* were assembled from the soil metagenomes. All three AMCCs exhibited relatively greater abundances in soils under relatively high As stress than in soils under low As stress (average of 0.52 vs. 0.13 copies per cell; Fig. 3c). Moreover, the transcriptional activities of these AMCCs in soils with relatively high As stress (average of 0.44) were also greater than in soils with low As stress (average of 0.02; Fig. 3d). The even distribution of metatranscriptomic reads aligned to the associated ARGs or MRGs in the AMCCs, as revealed by a read recruitment plot (Fig. 3e), confirmed that these ARGs and MRGs were expressed similarly in the soil microbial genome and likely confer resistance to both antibiotics and As.

### Factors contributing to the coexistence of ARG-MRG in agricultural soil microbiome
The partial least square-structural equation model (PLS-SEM) was used to describe the direct and indirect impacts of various factors on the coexistence of ARG-MRG (AMCC abundance). A goodness of fit (GOF) of 0.55 was achieved for the PLS-SEM and could explain 41% of the variance in the coexistence of ARG-MRG in the soil microbiome (Fig. 4a, b). Similarly, fertilizer type; soil properties, including pH, organic carbon content, and density; climate; and socioeconomic factors, including mean annual temperature (MAT), annual precipitation (AP), and the human development index (HDI), had the significant contributions ($p < 0.05$). Biofactors, including the relative abundances of ARGs, MRGs, and potassium metabolism genes, were also significantly important influencing factors based on the random forest model (Fig.4c). The fertilizer type indirectly impacts the coexistence of ARG-MRG in the soil microbiome by directly affecting the soil properties and the relative abundances of ARGs and MRGs. Climate and socioeconomic factors also contributed to the coexistence of ARG-MRG through indirect impacts on soil properties, microbial community diversity, and elemental metabolism gene abundances or through direct impacts on the abundances of ARGs and MRGs. Interestingly, climate contributed to the coexistence of ARG-MRG in the microbiome, with greater positive impacts on the NOF than OF soils (total effect of 0.53 vs. −0.14), while the biofactors, i.e., the abundance of ARGs and MRGs, were the predominant factors contributing to the coexistence of ARG-MRG in the OF soil microbiome (Supplementary Fig. 8).

### Identification of ARG-MRG-carrying MAGs
Approximately 15% (95 out of 617 in total) of the total metagenome-assembled genomes (MAGs) with relatively high quality (completeness

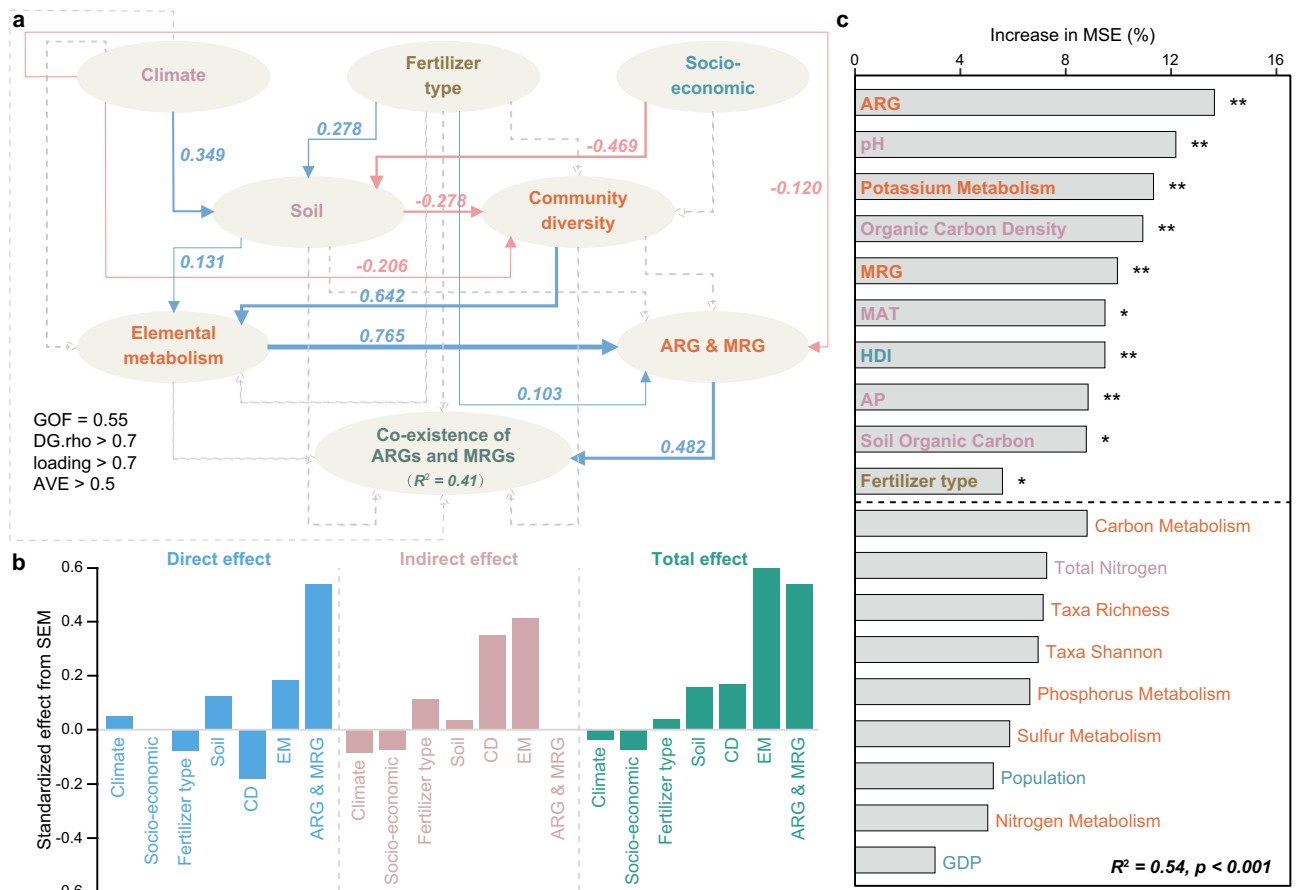

**Fig. 4 | Geographical variables affecting the abundances of coexistent ARG-MRGs in agricultural soil microbiomes. a** Partial least square-structural equation modeling (PLS-SEM) illustrating the direct and indirect effects on the coexistence of ARG-MRG (AMCC abundance). Blue, red, and gray indicate significantly positive, significantly negative, and nonsignificant effects, respectively. Path coefficients and coefficients of determination ($R^2$) were calculated after bootstrapping ($n = 100$), and all path coefficients shown were statistically significant ($p < 0.05$). Different colors represented different types of variables. **b** Standardized direct (blue),

indirect (pink), and total effects (green) from the PLS-SEM. **c** Mean predictor importance of different factors (same as PLS-SEM) on the abundance of AMCCs based on random forest modeling (Permutation test, $n = 1000$, *$p < 0.05$; **$p < 0.01$). Source data were provided as a Source Data file. ARG antibiotic resistance gene, MRG metal(loid) resistance gene, GOF goodness of fit, DG.rho Dillon-Goldstein's rho, AVE average variance extracted, CD community diversity, EM elemental metabolism, MAT mean annual temperature, AP annual precipitation, GDP gross domestic product, HDI human development index.

≥ 70% and contamination ≤ 5%) were identified as MAGs carrying AMCCs. Moreover, these MAGs were all detected with MGEs and VFGs in their genomes (Fig. 5a), indicating the potential risk of HGT in ARG-MRG co-resistant and pathogenicity in agricultural ecosystems. Among the antibiotic and metal(loid) co-resistant bacteria (AMRB), these MAGs were mostly resistant to antibiotics, including multidrug, fosmidomycin, bacitracin, etc., and metal(loid) including Cr, As, Fe, multimetal, Cu, Zn and Se (Fig. 5a), which is consistent with the dominant ARG and MRG types revealed based on the analyses of metagenomic reads and AMCCs (detailed in the Supplementary Results). According to the Microbial Genomes Atlas (MiGA) classification, three of these AMRB represented a potentially new genus with a probability of $p < 0.01$, 72 represented a potentially new species ($p < 0.01$), and 17 represented a potentially new subspecies ($p < 0.001$; Supplementary Table 6), indicating the novelty of these newly identified AMRB in agricultural soils. The total abundance of these AMRB was consistently greater in OF than NOF soils (0.011 *vs.* 0.005 CPG; Fig. 5b). While the dominant AMRB were composed of *Gammaproteobacteria* (44%), *Bacteroidia* (20%), and *Actinomycetia* (15%) in OF soils, in NOF soils, the AMRB were predominantly assigned to *Acidobacteriae* (41%), *Nitrospiria* (18%) and *Gammaproteobacteria* (16%; Fig. 5c and Supplementary Fig. 9). Among them, the observed MAGs, i.e., OF_MAG49 (*Enterobacter*), OF_MAG23 (*Pseudomonas*), and OF_MAG51

(*Pseudomonas*), which had the highest number of VFGs, are typical pathogenic genera confirmed by previous studies[21].

## Prediction of the global threat potential of AMRB in agricultural soils based on machine learning

Five machine learning (ML) algorithms (artificial neural network, k-nearest neighbors, support vector machine, extreme gradient boosting, and random forest) were further used to map the threat levels of AMRB in agricultural lands worldwide with 511 collected soil metagenomes and 34 features from public databases. The abundance of AMRB in 511 agricultural soils was divided into six levels by the K-means algorithm, with the rank six to one indicating the relative abundance of AMRB and their associated potential risk from high to low (detailed in Supplementary Results, Supplementary Fig. 10). Subsequently, the pre-processed dataset was divided into a training set (80%) and a test set (20%). The training set was used to train the model to predict the risk level of AMRB with 10-fold cross-validation and the test set was used to evaluate the model performance. The test set was never evaluated by the model until the final model was reached. After hyperparameter optimization and feature selection, the best model of each ML algorithm was validated on the same test set. Finally, the best model (feature selection: 23, n estimators: 50, max depth: 27, max features: None, min samples split: 7, and min samples leaf: 1) of the

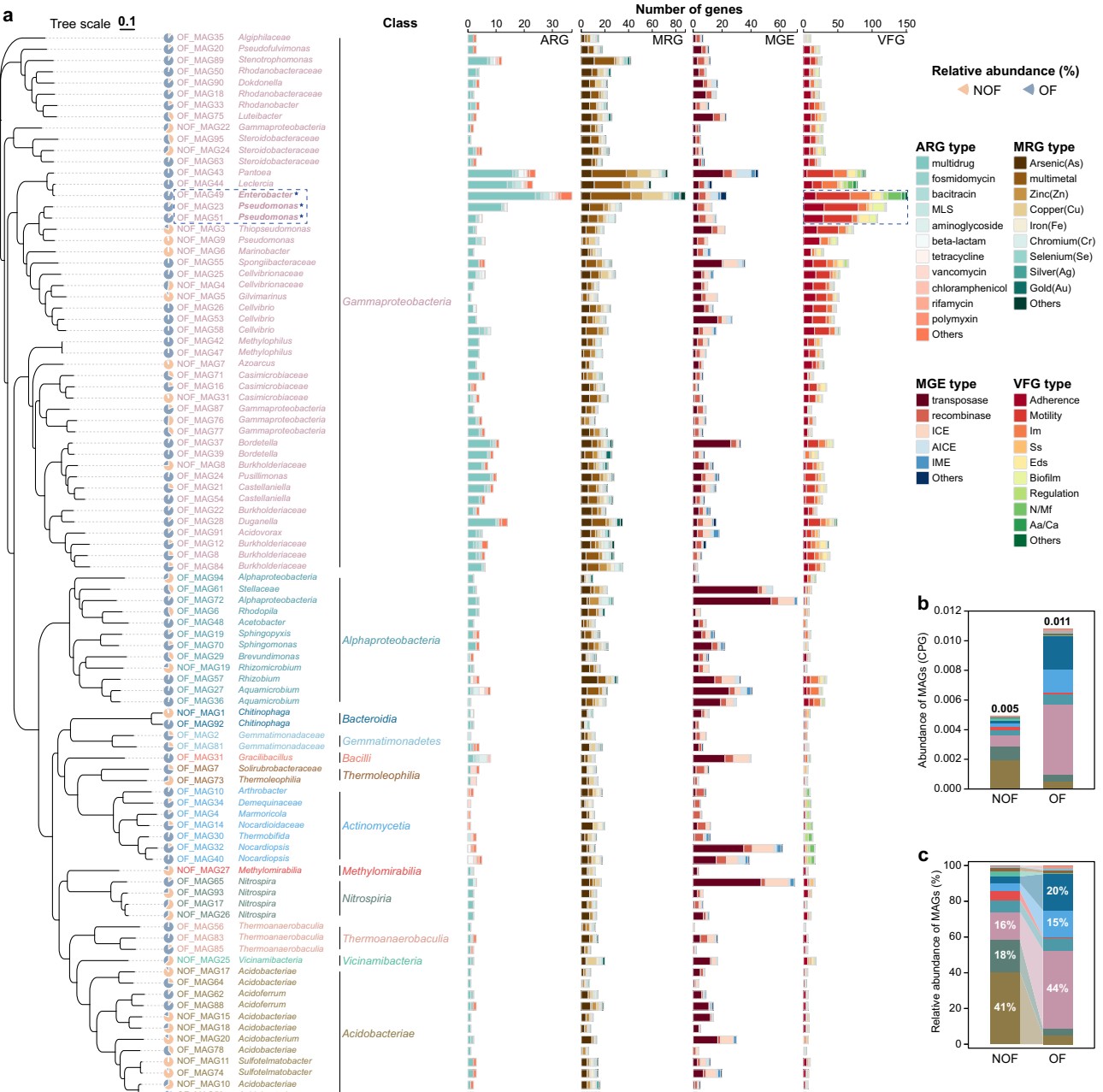

**Fig. 5 | Characterization of ARG-MRG-carrying metagenome-assembled genomes (MAGs). a** Phylogenetic tree of ARG-MRG-carrying MAGs (completeness ≥ 70% and contamination ≤ 5%). The text represented the taxonomic classification of each MAG at the Genus level or Family level; otherwise, the MAG was not classified to the Genus level. The colors of MAGs represented the different Class levels to which these MAGs were assigned. Pie charts represent the relative abundance of each MAG in the NOF and OF soils. The numbers of identified ARGs, MRGs, MGEs, and VFGs among the MAGs were shown in the bar plots. **b** Abundance of different host-MAGs (antibiotic and metal co-resistant bacteria, AMRB) at the Class level in the NOF and OF soils was calculated by the average coverage of all contigs in the

MAG and normalized by per genome equivalents (CPG). **c** Relative proportions of different host-MAGs (AMRB) at the Class level in the NOF and OF soils. In **b**, **c** different colors represented different Class levels, consistent with (**a**). Source data were provided as a Source Data file. ARG antibiotic resistance gene, MRG metal(loid) resistance gene, MGE mobile genetic element, VFG virulence factor genes, MLS macrolide-lincosamide-streptogramin, Im immune modulation, Ss stress survival, Eds effector delivery system, N/Mf nutritional/metabolic factor, Aa/Ca Antimicrobial activity/Competitive advantage, MAG metagenome-assembled genome.

random forest algorithm that performed best on the test set was used to predict the threat level of AMRB (F1 score of 0.722; Fig. 6a, b). The receiver operating characteristic curve (ROC) for each level further evaluated the performance of the random forest model (Fig. 6c).

After determining the best predictive model, we extracted the latitudinal and longitudinal coordinates of the global croplands based on Google Earth Engine (GEE) data, and ultimately, the microbial risk at 2,910,454 locations was predicted and visualized at a 0.083° of

resolution on the global map of agricultural soils (Fig. 6d). This map could provide an overview of AMRB in agricultural soils worldwide. Considering the multiple stresses in agricultural soils under fertilization practices, microbes conferring both antibiotic and metal(loid) resistance would have advantages over either ARG or MRG-resistant microbes and could be the predominant microbes facilitating ARG spread and resulting in potential risk. According to the prediction results from machine learning, a relatively higher risk of AMRB was

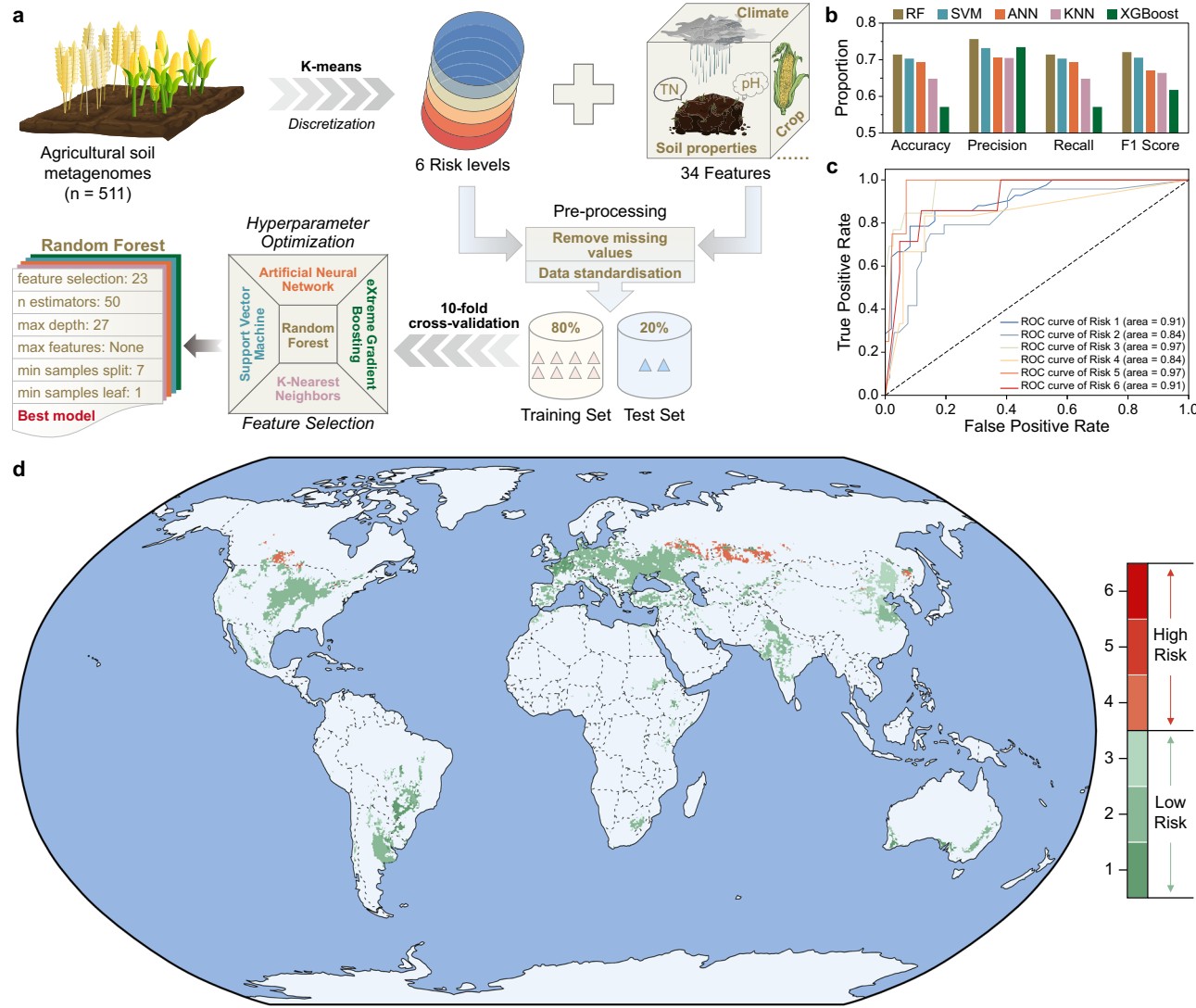

**Fig. 6 | Global mapping of the risk of antibiotic and metal(loid) co-resistant bacteria (AMRB) in agricultural lands. a** Process of building machine learning models. **b** Evaluation metrics (accuracy, precision, recall, and F1 score) for the best model of each machine learning algorithm on the test set. **c** Receiver operating characteristic (ROC) curve reflected the classification performance of the best model of random forest model on the test set at different risk levels. **d** Risk of the antibiotic and metal(loid) co-resistant bacteria (AMRB) in agricultural soils according to machine learning prediction results displayed at a 0.083° resolution using Python. The unsupervised learning approach using k-means clustering was applied to categorize the abundance of AMRB in 511 agricultural soils into six risk levels, which were visualized by the t-distributed stochastic neighbor embedding (t-SNE) method. According to the t-SNE results, a significant gap separates the samples into two groups, in which risk levels of 1, 2, and 3 indicate the low-risk group, and the risk levels of 4, 5, and 6 indicate the high-risk group. Source data were provided as a Source Data file. RF random forest, SVM support vector machine, ANN artificial neural network, KNN K-nearest neighbors, XGBoost: eXtreme Gradient Boosting, ROC receiver operating characteristic.

observed in central North America, Eastern Europe, Western Asia, and Northeast China (Fig. 6d).

## Discussion

The application of organic fertilizers, especially manure, can directly import resistance genes[22–24], and thus drive the spread of diverse ARGs and MRGs in agricultural soils worldwide (Fig. 1). The antibiotics and metal(loid) introduced during manure fertilization in agroecosystems[5,15] could also drive the co-selection of antibiotics and heavy metal(loid) resistomes observed here and elsewhere[8]. While previous studies have shown the co-occurrence of ARGs and MRGs in organic fertilizer agricultural soils based on the correlation between their abundances, we further reported a greater abundance of genetic linkages of ARG-MRGs in AMCCs in OF than NOF soils (Fig. 2). In addition to the genetic linkage of ARG-MRGs reported in isolated genomes[11,13,16], our results confirmed that these

linkages can be found within complex microbial communities occupying agricultural soils, such as *Gammaproteobacteria*, *Bacteroidia*, and *Actinomycetia* (Fig. 5). Specifically, the coexistence of *mdtA*, *mdtB* and *mdtC* which are involved in the cross-resistance mechanism of microbiomes[25,26], was the most common AMCC types in agricultural soils. Moreover, with organic fertilizer application, an increased proportion of AMCC types associated with co-resistance mechanisms in the soil microbiome[13,27,28] was observed, including multidrug-multimetal, multidrug-Cd, As, Cu, Se, Au, and Fe co-resistance. This co-resistance could be selected by the exogenous Cu, Zn, and As through the application of animal manure in soils, which has been used as a feed additive in food animals[5,29–31]. The greater abundance and transcriptional activity of ARG and MRG on AMCCs under higher As stress than under lower As stress (Fig. 3), further confirmed that the ARG and MRG were co-selected and co-regulated under stress in agricultural soils.

Soilborne ARGs have been recognized as ancient products of evolution[32,33] and essential microbial warfare for competing for soil resources[34]. Nevertheless, in agricultural soils, the nutrient levels of soil C, N, P, and K are typically high due to fertilization practices, and bacterial communities are usually associated with increased diversity resulting from an increased niche availability[35]. Therefore, instead of acting as warfare for competing for soil nutrients, the presence of resistomes in this habitat is more likely to aid in resisting the stress of antibiotics and metal(loid) presented. At the microbial community level, land-use perturbations have been identified as the most important anthropogenic pressures affecting soil microbial diversity[36,37]. With the application of organic fertilizers, the pressure from antibiotics and metal(loid) reduced microbial diversity at the community level (Fig. 1b) and selected the resistomes conferring both antibiotic and metal(loid) resistance (Figs. 2 and 3), such as *Pseudomonadota*, *Actinobacteriota*, and *Acidobacteriota* (Supplementary Fig. 9), which have been identified as ARG hosts[38,39] or with high tolerances of metal(loid)[40]. Recently, with an increasing awareness of antibiotic-resistant pathogens embedded within complex species interaction networks, ecological coexistence theory has been utilized to understand antibiotic resistance[41]. We suspected that the enhanced microbial interaction (Fig. 1) in organic fertilized agricultural soils would contribute to the co-resistance of the microbial communities to the presented stresses through possible HGT of the resistance genes, as a significantly greater abundance and variant types of MGEs were revealed on the AMCCs in OF soils (Fig. 2) and in previously studied manure-amended soils[23]. Corroboratively, HGT of ARGs through different strains has been confirmed to be an important way for transmission of resistance through microbial communities[42].

In contrast to the previously studied ARGs in various types of soil environments which could be contributed by complex factors, including both ancient evolution[32] and anthropogenic activities[43], we focused on agricultural ecosystems in which ARG and MRG abundances and co-selection were more strongly contributed by certain factors, such as fertilization types and associated soil properties or climate factors (Fig. 4). The contributions of these factors could be quantified more accurately and thus generate a more robust and reliable prediction model for AMRB in agricultural lands, promoting the understanding of the potential risk of AMRB at a global scale. A relatively greater risk of AMRB was revealed in agricultural soils with organic fertilizer applied (Fig. 5), especially in Central North America, Eastern Europe, Western Asia, and Northeast China (Fig. 6). A similar distribution pattern of soil ARG hot spots has been revealed in these locations[44] and was most likely resulted from the highly dense populations[45–47] and the associated human activities such as livestock, crop production, irrigation, manure application, and barley and sheep production, which contributed to the spread of ARGs[44]. The relatively greater abundance of ARGs presented in the soils located in these areas could also contribute to a greater abundance of resistomes conferring both antibiotic and metal(loid) resistance, considering that genes conferring resistance to antibiotics and other contaminants are usually carried by the same MGEs and were co-selected[48,49].

In conclusion, our study revealed the worldwide coexistence of ARG-MRG in agricultural soils based on the genetic linkage evidence from global agricultural soil metagenomes and indicated that organic fertilization facilitates the coexistence of ARG-MRG in the agricultural soil microbiome. In addition to the coexistence patterns of ARG-MRG, the results of the metatranscriptomic data confirmed the increase in the co-transcript levels of ARG and MRG under As stress, further advancing our understanding of the co-selection of soil resistomes. We also constructed a global spatial distribution map of AMRB in agricultural lands based on machine learning and propose that the co-resistant resistomes to both antibiotics and metal(loid) in agricultural soils should be brought to the forefront, especially considering their

genetic transmission potential and the intensive pressure from anthropogenic activities during cultivation practices. Our study highlights the impact of organic fertilization on the coexistence and potential dissemination of ARGs and MRGs in global agricultural soils, underscoring the need for further investigation to understand and mitigate the spread of these co-resistant genes to safeguard public health.

## Methods

### Metagenomic dataset collection
Metagenomic samples from agricultural soils worldwide were retrieved from the Sequence Read Archive (SRA, https://www.ncbi.nlm.nih.gov/sra) database, the European Nucleotide Archive (ENA, https://www.ebi.ac.uk/ena/browser/home), the DNA Data Bank of Japan (DDBJ, https://www.ddbj.nig.ac.jp), and the National Genomics Data Center (NGDC, https://ngdc.cncb.ac.cn/) by searching for the keyword 'agricultural soil' on 2022-10. The data obtained were subsequently refined using the following criteria: (1) Plant-associated samples, such as rhizosphere and rhizoplane soils, were excluded; (2) Only topsoil (depth < 20 cm) metagenomic samples were retained; (3) Only paired-end sequencing reads were generated by Illumina shotgun platforms were included; and (4) The data size (number of bp) of every metagenomic sample was greater than 1 Gb. After rigorous screening, a total of 511 agricultural soil metagenomes were collected.

### Classification of the agricultural soil groups using the random forest classification model
Based on the data uploaded or corresponding article information, 109 of the 511 global agricultural soil samples were classified as fertilized with organic fertilizer (OF), and 109 samples were classified as not fertilized with organic fertilizer (NOF). Due to insufficient information on fertilization types, 293 of the 511 agricultural soil samples lacked information on fertilization types. The trained random forest (RF) classification model was built based on the information of the 109 identified NOF samples and 109 OF samples by the 'randomForest' R package (v4.7-1.1) and was subsequently used to classify 293 agricultural soil samples as NOF or OF samples (detailed in the Supplementary Results). Briefly, after taxonomic classification (the methods are provided in the following paragraph), the abundances of all the genera in the 109 NOF and 109 OF samples were used to construct an RF classification model. In each group, 70% of the samples were randomly selected as the training set to train the classification model, and the rest were used as the test set. To ensure the representativeness of the training and test sets in each group, the data splitting strategy and ten-fold cross-validation were applied and detailed in the supplementary.

### Metagenomic assembly, open reading frame (ORF) prediction and binning
The raw data of 511 metagenomes were downloaded from Kingfisher (wwood.github.io/kingfisher-download). The raw reads were trimmed using fastp v0.22.0[50], after which the reads were removed by the Phred quality score (Q < 20), the number of ambiguous ≤ 3, and a minimum fragment read length of 50 bp after trimming for downstream analyses. The clean reads from each sample were individually assembled into contigs using MEGAHIT v1.2.9[51] with the parameters '-min-contig-len 500'. Metagenome-assembled genomes (MAGs) were recovered from the contigs longer than 1000 bp by MetaWRAP v1.2.1[52]. The resulting MAGs were improved by the 'Bin_refinement' model of MetaWRAP. The completeness and contamination of the refined MAGs were assessed using CheckM v1.1.3[53] and only those MAGs with completeness ≥ 50% and contamination ≤ 10% were included in the succeeding analysis. The MAGs were subsequently dereplicated using dRep v3.3.0[54] with the parameters '-sa 0.95 -nc 0.30'. The ORFs were

predicted from the contigs of each sample or MAG by Prodigal v2.6.3[55] with the parameter '-p meta'. The taxonomic affiliation of the MAGs was determined by the 'classify_wf' model of GTDB-Tk v2.1.0[56]. Phylogenetic analysis of MAGs was conducted with the 'infer' module of GTDB-Tk, and the phylogenetic tree was visualized in tvBOT v2.5.0[57]. The taxonomic novelty of the MAGs was analyzed with the Microbial Genomes Atlas (MiGA)[58]. The abundance of the MAG was calculated in a few steps. First, clean reads from each sample were mapped to the contigs of the MAG using BLAT v2.3.4.1[59] with an e-value less than 1e-5, at least 95% identity, and at least 70% query coverage, and only the top hits were retained. Next, a script (http://enve-omics.ce.gatech.edu/enveomics/docs?t=BlastTab.seqdepth.pl) was used to calculate the coverage of each contig in the MAG. The average coverage of all contigs in the MAG was used as the coverage of the MAG. The genome equivalents, which were equal to the total bp sequenced/average genome size in bp, for each sample were calculated by MicrobeCensus v1.1.0[60]. Finally, the abundance of the MAG was normalized to that of the CPG (coverage per genome equivalent).

### Taxonomic classification and functional gene annotation based on clean reads

Taxonomic classification of metagenomes was conducted using Kraken2 v2.07[61] with the parameter '-c 0.05', and the relative abundance was estimated using Bracken v2.7[62]. To annotate more comprehensive MRG and MGE information, we constructed more complete MRG and MGE databases. The self-constructed MRG database consists of part of the BacMet database with experimentally confirmed for MRGs[63], the copper resistance protein database[64], and As resistance genes from UniProt. The self-constructed MGE databases used included ICEberg[65], INTEGRALL[66], ISfinder[67], nanoMGE[68], and The Transposon Registry database[69]. All the clean reads were searched against the structured ARG reference database (SARG), the self-constructed MRG database, the self-constructed MGE database, and the virulence factor database (VFDB) to annotate the potential ARGs, MRGs, MGEs, and VFGs, respectively. BLASTx of DIAMOND v2.0.14.152[70] was used for the alignment of sequences at an e-value cutoff of 1e-7, and only the top BLAST hit was retained for further filtration. A sequence was annotated as an ARG/MRG/MGE/VFG-like fragment if it met the following criteria: ≥ 90% identity and an alignment length ≥ 25 aa[71]. Additionally, Rank I and Rank II ARGs were considered risk ARGs according to previously reported standards[72]. Similarly, functional genes were annotated by the SEED database[73] by best-hit classification with a maximum e-value of 1e-5, a minimum identity of 60%, and a minimum alignment length of 25 aa[74]. The abundances of ARG/MRG/MGE/VFG/functional genes were normalized as RPKG = (reads mapped to gene)/(gene length in kb)/(genome equivalents), which indicates the fraction of total cells encoding the gene of interest, i.e., copies per cell[75].

### Co-occurrence patterns of microbes

Microbial co-occurrence networks at the genus level were constructed using the 'RMThresh' R package. To avoid potential spurious associations of rare genera affecting reliability, the data were filtered prior to correlation calculations. The genera present in 80% of the NOF or OF soils respectively, were selected for network construction based on Spearman correlations of their relative abundances, which were determined by the random matrix theory (RMT)-based approach. Topological properties, including the number of nodes and edges, relative modularity, connectance, average clustering coefficient, and average degree, were computed by the 'ggClusterNet' R package[76] and finally visualized by Gephi v0.9.2[77]. We calculated the robustness (i.e., the resistance to node loss) of the networks with 50% of the genera randomly removed from each network.

### Identification of the genetic linkage between ARGs and MRGs in contigs and MAGs

ORFs were annotated as potential ARGs, MRGs, MGEs, or VFGs using the BLASTp of DIAMOND[70] by searching against the four databases above, respectively. The ORFs from the contigs or MAGs were annotated as ARG/MGE/VFG-like ORFs according to the following criteria: minimal identity of 60%, minimum query coverage of 70%, and e-value below 1e-5. Only the top BLAST hit was retained. The MRG-like ORFs were identified with a minimal identity of 50%, minimum query coverage of 70%, and e-value below 1e-5. Contigs greater than 1000 bp in length were predicted to be plasmid or chromosome sequences using PlasFlow[78]. The contigs (> 1000 bp in length) carrying at least one ARG plus one MRG were considered ARG-MRG-carrying contigs (AMCCs). The relative abundances of AMCCs in each sample were calculated as the sum of the relative abundance of the ARG/MRG-like ORFs annotated on AMCCs [(reads mapped on ORF)/(ORF length in kb)/(genome equivalents)] divided by the total number of identified ARG/MRG-like ORFs on the AMCCs, which indicated the fraction of total cells encoding the AMCC. The coexistence pattern of the ARG and MRG subtypes was displayed by a network constructed using Gephi.

### Metagenomic and metatranscriptomic analyses of paddy soil samples under different As stresses

Soil samples were collected from four different paddy fields in China in July 2022, including three samples in Yuchangping (CS_YCP_1/2/3), three samples in Huaihua (HN_HH_1/2/3), three samples in Baiyun (CD_BY_1/2/3), and three samples in Qujiang (ZJ_QJ_1/2/3). Paddy soil samples were placed in sterile plastic bags and transported to the laboratory on ice for DNA and RNA extraction and soil physiochemistry analysis. Soil properties, including pH, the total concentration of organic carbon (TOC), nitrogen (TN), phosphorus (TP), sulfur (TS), and As were determined on a composite sample of the three replicates following standard methods[79]. Soil DNA and total RNA were extracted from soil using the RNeasy PowerSoil Total RNA Kit (Qiagen) and RNeasy PowerSoil DNA Elution Kit (Qiagen), respectively, according to the manufacturer's protocol. The DNA and RNA extracted from each sample were subsequently sent to Majorbio Bio-Pharm Technology Co., Ltd. (Shanghai, China) for metagenome sequencing and metatranscriptome sequencing, respectively.

Metagenomic raw data from 12 samples were processed using the same methods and parameters as described above, including quality control, metagenomic assembly, ORF prediction and annotation, and relative abundance of AMCCs. The raw metatranscriptomics data of 12 samples were quality-controlled by default parameters of multitrim (https://github.com/KGerhardt/multitrim). SortMeRNA v4.3.4[80] was used with default settings to remove residual rRNA sequences after rRNA subtraction from the metatranscriptomes. To calculate the transcriptional activity of AMCCs obtained from 12 metagenomes, clean reads from the metagenomes and metatranscriptomes were mapped to AMCCs using BLAT v2.3.4.1 with an e-value below 1e-5, at least 90% identity and a query coverage ≥ 70%, respectively. The relative abundance of AMCCs in the metagenomes was calculated as described above. The transcriptional activity of the AMCCs was evaluated by the relative abundance of AMCCs in the metatranscriptomes [RPKM = (the number of clean reads mapped to the AMCC)/(the AMCC length in kb)/(the total number of clean reads)] normalized by the relative abundance of the same AMCCs identified in the metagenomes.

### Estimation of factors contributing to ARG-MRG coexistence using SEMs

To explore the effects of multiple variables on ARG-MRG coexistence, partial least square-structural equation modeling (PLS-SEM) was employed to explore the direct, indirect, and interactive effects ('plspm' R package). Climate variables, including mean annual temperature (MAT) and annual precipitation (AP), were obtained from

WorldClim (https://worldclim.org/data/worldclim21.html). Socio-economic variables, including population, gross domestic product (GDP), and the human development index (HDI), were obtained from https://data.worldbank.org.cn/. Soil variables, including pH, soil organic carbon, organic carbon density, and total nitrogen were obtained from SoilGrids (https://soilgrids.org/). The Shannon diversity index and Richness diversity index were used as microbial community diversity variables. The abundances of carbon, nitrogen, phosphorus, sulfur, and potassium metabolism genes were used as elemental metabolism variables. Detailed information on the latent variables in the PLS-SEM and the corresponding observable variables were listed in Supplementary Table 8. Indirect effects were defined as multiple path coefficients between the predictor and response variables including all possible paths excluding the direct effect[81]. The final model selection was based on Cronbach's alpha (C.alpha), Dillon-Goldstein's rho (DG. rho), loading, average variance extracted (AVE), and goodness of fit (GOF), which evaluate the model's reliability and goodness of fit.

### Machine learning algorithms

To generate a global predictive model of microbial risk associated with AMRB, we collected 34 features from public databases, including latitudinal and longitudinal data, climatic data, soil property data, crop yield data from EARTHSTAT, livestock density data from the Food and Agriculture Organization of the United Nations (FAO, https://www.fao.org/livestock-systems/global-distributions/en/), and HDI data. To improve the stability and accuracy of the machine learning model and avoid the influence of extreme values, the data were discretized using an unsupervised K-means algorithm. The abundance of high-risk MAGs from 511 samples of global agricultural soils was divided into 6 levels after discretization. The best model was selected according to the following steps. First, by removing the samples containing missing values and normalizing the data, we divided the dataset into training (80%) and testing (20%) sets. Second, the training set was used to train 5 machine learning (ML) algorithms (artificial neural network, k-nearest neighbors, support vector machine, extreme gradient boosting, and random forest) models separately and 10-fold cross-validation was added to avoid overfitting. Third, after hyperparameter optimization and feature selection, the best model was generated for each ML algorithm. The best models of the 5 different ML algorithms were validated on the same test set, and their accuracy, precision, recall, and F1 score were compared. Finally, the ML algorithm model that performed the best on the test set was selected for the final global prediction. The final predictions were visualized through the Matplotlib, Cartopy, Rasterio, and Numpy libraries of Python.

### Statistical analyses

The map of the geographic location of globally collected soil metagenomes was generated using the 'ggmap' R package (https://github.com/dkahle/ggmap). The alpha diversity of the microbial communities and functional genes (Shannon and Richness) was calculated by the 'amplicon' R package. The beta diversity of the microbial communities was assessed by Bray-Curtis similarity based on the species level of taxonomic classification or functional microbial communities classified at the subsystem level using the 'vegan' and 'dplyr' R packages. Spearman correlation analyses were performed with the 'ggcorrplot' and 'corrplot' R packages. Linear regression analyses and plotting were performed with the 'ggpubr' and 'ggpmisc' R packages. A Sankey graph was constructed with the 'tidyverse' and 'networkD3' R packages. Two-group comparisons were analyzed by the two-sided Wilcoxon test ('dplyr' R package) or DESeq2 analysis ('DESeq2' R package).

### Reporting summary

Further information on research design is available in the Nature Portfolio Reporting Summary linked to this article.

## Data availability

The accession codes for the 511 collected global soil metagenomes are available in Supplementary Table 1. The 12 metagenomic and meta-transcriptomic raw sequencing data generated in this study have been deposited in the National Center for Biotechnology Information (NCBI) SRA database (https://www.ncbi.nlm.nih.gov/sra) under the BioProject PRJNA1068274 and PRJNA1068685, respectively. Source data are provided with this paper (https://doi.org/10.6084/m9.figshare.25144382).

## Code availability

All the scripts and codes for classifying samples and machine learning used in this study were available online at Code Ocean (https://doi.org/10.24433/CO.8910377.v1).

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

## Acknowledgements

This research was funded by the National Natural Science Foundation of China (52270198).

## Author contributions

S.Z. and Z.L. designed this research and wrote the original draft of the manuscript. Z.L. and R.M. contributed to the data curation. Z.L. performed the data analyses. S.Z. provided the supervision and funding acquisition. K.K., D.Z., and Y.Z. performed the conceptualization, manuscript reviewing, and language modification.

## Competing interests

The authors declare no competing interests.
