## [Peer Review File · Nature Communications]

REVIEWER COMMENTS

Reviewer #1 (Remarks to the Author):

General Comments

This study investigated the impact of organic fertilizer application on the coexistence of antibiotic (ARGs) and heavy metal resistance genes (MRGs) through an extensive literature review. Research findings elucidated the promotive effect of organic fertilization on the coexistence of ARG-MRG in agricultural soil. Concurrently, transcriptomic data validation indicated an increased co-transcription level of ARG and MRG under As stress. Finally, a global distribution map of AMRP was constructed using machine learning. However, there are some detailed lapses in the expression of the article content. The following points are provided to hopefully improve the manuscript.

Specific Comments:

1. Lines 21-34, certainly, it is advisable to provide additional elucidation on specific key data points within the abstract. Incorporating numerical values or percentages related to the observed threefold increase in diverse types of ARG-MRG-carrying contigs (AMCCs), as well as quantifying the heightened relative abundance in the mentioned regions, would enhance the precision and comprehensiveness of the presented information.
2. Lines 39, prioritize the uniformity of paragraph formatting.
3. Lines 70-71, perhaps it would be feasible to elaborate on the genetic linkages observed?
4. Line 80, why choose As as the primary research focus? It might be worthwhile to briefly elaborate on the reasons for this choice at this juncture.
5. Lines 156-169, this study addresses a global-scale issue, and conducting validation experiments using soil samples from China ensures global representativeness, doesn't it? Furthermore, how are the standards defined for high and low concentrations of As stress?

6. Line 197, why is Se recognized as one of the metal elements? The basis for this classification originates from where?

7. Lines 335-337, what measures were taken to ensure the representativeness of the training and test sets in each group?

8. Lines 299-309, in conclusion, incorporating sentences that emphasize the significance of the research at the end of the conclusion could provide a comprehensive summary.

9. Lines 515-710, the references contain errors, please correct them.

10. Fig 5, given the extensive information intended to be conveyed in the figures, it is preferable to utilize annotations to highlight key points for emphasis and clarity of expression.

11. Fig 6, in terms of the criteria for defining high and low risk, what standards were employed? It would be preferable to articulate this information in the figure caption for reader convenience.

Reviewer #2 (Remarks to the Author):

The authors' analysis of 511 global agricultural soil metagenomes shows that organic fertilization significantly increases the variety and quantity of groups carrying antibiotic and metal resistance genes compared to non-organic soils. This fertilization also affects the co-occurrence and expression of these genes, especially under arsenic stress, impacting soil properties and gene abundances. Moreover, through machine learning, the study investigated the distribution and potential pathogenic risks of these genes in agricultural lands, ultimately identifying high-risk areas in Central North America, Eastern Europe, Western Asia, and Northeast China.

This is a very interesting study. I really enjoy reading it. The figures are well presented, and the text is mostly written in a way that it is easy to follow for the reader (exceptions below). I have a few minor comments before I can recommend the paper for publication.

Minor comments:

The pipelines of each analysis are not provided.

Line 22: "Antibiotic (ARGs)" should be "Antibiotic Resistance Genes (ARGs)".

There is no reference to Fig. 2c in the manuscript.

Fig. 5c: the "Relative abundanceof MAGs (%)" should be "Relative abundance of MAGs (%)".

Reviewer #1 (Remarks to the Author):

General Comments:

This study investigated the impact of organic fertilizer application on the coexistence of antibiotic (ARGs) and heavy metal resistance genes (MRGs) through an extensive literature review. Research findings elucidated the promotive effect of organic fertilization on the coexistence of ARG-MRG in agricultural soil. Concurrently, transcriptomic data validation indicated an increased co-transcription level of ARG and MRG under As stress. Finally, a global distribution map of AMRP was constructed using machine learning. However, there are some detailed lapses in the expression of the article content. The following points are provided to hopefully improve the manuscript.

Response: Thank you very much for your comments. We have addressed the specific comments one by one below.

Specific Comments:

1. Lines 21-34, certainly, it is advisable to provide additional elucidation on specific key data points within the abstract. Incorporating numerical values or percentages related to the observed threefold increase in diverse types of ARG-MRG-carrying contigs (AMCCs), as well as quantifying the heightened relative abundance in the mentioned regions, would enhance the precision and comprehensiveness of the presented information.

Response: Thank you for your valuable suggestion, which enhanced the precision and comprehensiveness of our presented information in the abstract. We have incorporated the numerical values for the threefold increase in diverse types of AMCCs in lines 27-29 as “organic fertilization correlated with a threefold increase in diverse types of ARG-MRG-carrying contigs (AMCCs) in the microbiome (63 types) compared to non-organic fertilized soils (22 types)”.

For the heightened relative abundance in the mentioned regions, we have revised it according to your suggestion as “The map unveiled a heightened relative abundance and potential pathogenicity risks (range of 4-6) for spreading of co-existent ARG-MRGs in Central North America, Eastern Europe, Western Asia, and Northeast China compared to other regions, which acquired a risk range of 1-3” in lines 35-38.

2. Lines 39, prioritize the uniformity of paragraph formatting.

Response: Thank you. We have formatted the first paragraph of the introduction to keep it consistent with the other paragraphs in line 48.

3. Lines 70-71, perhaps it would be feasible to elaborate on the genetic linkages observed?

Response: Thank you for your suggestion. We have elaborated the genetic linkages as “Genetic linkages have been previously observed based on bacterial isolate genomes, including co-existence of Zn and beta-lactam resistance, bacitracin and polymyxin resistance, and cadmium (Cd) and aminoglycoside resistance,” in lines 85-86.

4. Line 80, why choose As as the primary research focus? It might be worthwhile to briefly elaborate on the reasons for this choice at this juncture.

Response: Arsenic is one of the most commonly used food additives in livestock, for example, roxarsone is used as an additive to treat parasitic diseases and for animal fattening and has been reported to be transferred into the agricultural soils through manure application^{1, 2}. Moreover, As contamination is one of the most seriously reported types of metal(loid) contamination in agricultural soils and poses a health risk to people who consume rice worldwide³⁻⁵. Taking these into consideration, As was selected for further investigation.

Thank you for your suggestion. We have elaborated on the reasons for this choice in lines 95-112 as follows: “We also sampled 12 agricultural soils under different levels of metalloid treatment for metagenomic and metatranscriptomic analyses. Considering that the As organic compound roxarsone was one of the most commonly used food additives in livestock to treat parasitic diseases and for animal fattening^{1, 2}, as well as the worldwide reported As contamination in agricultural soils resulting from either manure application or geologic origins⁶, the stress of As on the transcriptional activities of the ARGs and MRGs found in the same DNA molecule and/or genome was further assessed”.

5. Lines 156-169, this study addresses a global-scale issue, and conducting validation experiments using soil samples from China ensures global representativeness, doesn't it? Furthermore, how are the standards defined for high and low concentrations of As stress?

Response: According to the global scale investigation, increased ARG-MRG-carrying contigs (AMCCs) were revealed to be correlated with organic fertilization. The AMCC types are mostly associated with the cross-resistance mechanism of microbiomes, that is, carrying the *mdt* genes conferring resistance to both metal(loid)s and antibiotics. This cross-resistance mechanism has been well documented in a previous study⁷ showing the upregulation of the expression of the *mdt* genes involved in the efflux of Zn and multidrug. The different MRGs and ARGs carried by the remaining AMCCs suggested the co-resistance mechanisms in the soil microbiome. Among them, we identified a gene cluster (plasmid) with almost identical genes (with identity and query breadth coverage by amino acids greater than 90%) of arsenic resistant (*acr3*, *arsC2*, *arsH*, and *arsR3*), transposon (*Tn6183*, and *Tn6302*) and sulfonamide resistant (*sul2*) plants adjacent to each other in different OF soils at the global scale, suggesting their relative prevalence in agricultural soils. However, whether the As resistance and antibiotic resistance genes are co-regulated to confer the resistance remains unclear. To validate the co-resistance mechanism of this AMCC type, we selected agricultural soils under As stress for further investigation. The soil samples were collected from the agricultural soils in China as representative in this case to validate the co-resistance mechanism. Considering that the long-term application of organic fertilizers has also been reported to increase the concentration of As in agricultural soils⁸⁻¹⁰, we suspected that the stress from organic fertilizers would co-select for ARGs and MRGs. Accordingly, our results based on metagenomic and metatranscriptomic analyses of Chinese soils could provide robust support for the co-selection and co-regulation of ARGs and MRGs under As stress in agricultural soil. Nevertheless, various MRGs can co-exist with ARGs in addition to the As resistance genes, especially considering the heterogeneity of soils globally. Thank you for your comment. To address this, we also added a discussion regarding the need for further investigation in lines 356-357 as “underscoring the need for further investigation to understand and mitigate the spread of these co-resistant genes”.

For the standards defined for the stress of As, we used the National Soil Environmental Quality Standard of China (15 mg kg⁻¹ GB-15168-1995) to classify the soils into As-contaminated (average of 19.49 mg kg⁻¹) and As-noncontaminated (average of 9.32 mg kg⁻¹) soils. The microbiome in As-contaminated soils was subjected to under high As stress, and the microbiome in As-noncontaminated soils was considered to be under low As stress. Thank you for your comment. We have added details in lines 189-192 as follows: “According to the National Soil Environmental Quality Standard of China, which suggested soil contamination of As if the total soil As content was higher than 15 mg kg⁻¹, these soil samples were classified

into with relatively low levels of As stress (average As concentration of 9.32 mg kg⁻¹) and relatively high levels of As stress (average As concentration of 19.49 mg kg⁻¹) groups”.

6. Line 197, why is Se recognized as one of the metal elements? The basis for this classification originates from where?

Response: Thank you for the comment. Se is not a metal; it is actually a metalloid element. To improve the precision of the manuscript, we have carefully checked the manuscript thoroughly and changed ‘metal’ to ‘metal(loid)’ instead.

7. Lines 335-337, what measures were taken to ensure the representativeness of the training and test sets in each group?

Response: The measures taken to ensure the representativeness of the training and test sets including data splitting strategy and ten-fold cross-validation. Details of this information have been provided in the supplementary result section “Details of the trained Random Forest classification model”. Thank you for your comments. We also highlighted this information in the manuscript in the lines 389-391 as “To ensure the representativeness of the training and test sets in each group, the data splitting strategy and ten-fold cross-validation were applied and detailed in the supplementary”.

Specifically, the following measures were implemented: 1) Data Splitting Strategy: The data used for training were evenly distributed across the 2 groups (NOF:109 samples; OF:109 samples), which ensured that the proportions and samples of each subgroup were consistent in both the training and test sets, which improved the representativeness of the groups. 2) Cross-Validation: Most importantly, to mitigate the potential bias that can arise from a single division of data, we conducted ten-fold cross-validation with 10 repetitions. This approach not only reinforces the stability and reliability of the model's performance estimates but also guarantees that every sample in the dataset has an equal chance of being included in the test set, reinforcing the representativeness of our training and testing regimes. In summary, these measures (especially ten-fold cross-validation) collectively ensure the reliability and generalizability of our model, thereby ensuring the representativeness of the training and test sets across each group. The model's performance on the test set, with an accuracy of 0.97, a recall of 0.94, and a precision of 1, further attests to its predictive capability.

8. Lines 299-309, in conclusion, incorporating sentences that emphasize the significance of the research at the end of the conclusion could provide a comprehensive summary.

Response: Thank you for your constructive suggestion. We have incorporated sentences that emphasize the significance of the research at the end of the conclusion as “Our study highlights the impact of organic fertilization on the co-existence and potential dissemination of ARGs and MRGs in global agricultural soils, underscoring the need for further investigation to understand and mitigate the spread of these co-resistant genes and safeguard public health” in lines 354-357.

9. Lines 515-710, the references contain errors, please correct them.

Response: Thank you. We have revised the format of the references according to the journal’s guidelines. Corrections have been made to the references on lines 574-759. Additionally, we have updated all journal titles to their correct abbreviated format in accordance with the journal's referencing style and re-imported the references using Endnote to guarantee formatting consistency. Duplicate references were also identified and subsequently removed.

10. Fig 5, given the extensive information intended to be conveyed in the figures, it is preferable to utilize annotations to highlight key points for emphasis and clarity of expression.

Response: Thank you for your constructive suggestion. We have added annotations to highlight the key points for emphasis and clarity of expression (line 806). Specifically, the following changes have been made in Figure 5: 1) We have outlined the stacked bar graphs in Figure 5a with gray borders to improve the distinction between different gene types for the readers. 2) We have marked OF_MAG49, OF_MAG23 and OF_MAG with a blue border and a five-pointed star, these are the MAGs that carry the greatest number of VFGs. 3) In Figure 5b we have highlighted the abundance of MAGs in the NOF and OF samples. 4) In Figure 5c, we have emphasized the relative abundance of major taxonomic groups within MAGs in the NOF and OF samples.

11. Fig 6, in terms of the criteria for defining high and low risk, what standards were employed? It would be preferable to articulate this information in the figure caption for reader convenience.

Response: We employed an unsupervised learning approach using k-means clustering to categorize the abundance of antibiotic and metal(loid) co-resistant bacteria (AMRB) in 511 agricultural soils into six risk levels. The t-distributed stochastic neighbor embedding (t-SNE) method was applied to visualize the clustering results. According to the t-SNE results, a significant gap separated the samples into two groups, that is the low-risk group (risk levels 1, 2, and 3) and the high-risk group (risk levels 4, 5, and 6; Fig. S10). Regarding the abundance of AMRB, the AMRB abundance in samples categorized into risk levels of 1, 2, and 3 was between $3e-4$ to $9e-3$ coverage per genome equivalent (CPG). However, samples classified at risk levels of 4, 5, and 6 exhibited AMRB abundances ranging from $1e-2$ to $1e-1$ CPG. Given the clustering results and substantial disparity in AMRB abundance between these groups, we defined samples within risk levels of 1 to 3 as low risk and those within risk levels of 4 to 6 as high risk. The six risk levels therefore served as a critical indicator for differentiating between low- and high- risks. Upon defining the risk levels, a variety of features (such as climate data, soil properties) and machine learning algorithms were utilized to predict the microbial risk level for each point at a 0.083° resolution across global farmlands. The predicted risk levels of AMRB in different locations was used to create a map of microbial risk levels across global farmlands, where levels 1-3 were marked as ‘Low Risk’ areas and levels 4-6 were marked as ‘High Risk’ areas.

Thank you for your suggestion. We have articulated this information in the caption of Figure 6 as “The unsupervised learning approach using k-means clustering was applied to categorize the abundance of AMRB in 511 agricultural soils into six risk levels, which were visualized by the t-distributed stochastic neighbor embedding (t-SNE) method. According to the t-SNE results, a significant gap separates the samples into two groups, in which risk levels of 1, 2, and 3 indicate the low-risk group, and the risk levels of 4, 5, and 6 indicate the high-risk group” in lines 824-829.

We have also detailed the criteria for defining high and low risk in the supplementary results section, lines 278-298, and briefly introduced them in the manuscript, lines 256-257, as “with the rank six to one indicating the relative abundance of AMRB and its associated potential risk from high to low (detailed in the supplementary, Fig. S10).”

Fig. S10. Visualization of antibiotic and metal(loid) co-resistant bacteria (AMRB) abundance clusters in agricultural soil samples using t-SNE. Each point represents an individual soil sample, color-coded according to its assigned risk level based on the k-means method.

References

1. Huang, L.X. et al. Roxarsone and its metabolites in chicken manure significantly enhance the uptake of As species by vegetables. *Chemosphere* **100**, 57-62 (2014).
2. Fisher, D.J., Yonkos, L.T. & Staver, K.W. Environmental concerns of roxarsone in broiler poultry feed and litter in Maryland, USA. *Environ. Sci. Technol.* **49**, 1999-2012 (2015).
3. Kumarathilaka, P., Seneweera, S., Meharg, A. & Bundschuh, J. Arsenic speciation dynamics in paddy rice soil-water environment: sources, physico-chemical, and biological factors - A review. *Water Res.* **140**, 403-414 (2018).
4. Davis, M.A. et al. Assessment of human dietary exposure to arsenic through rice. *Sci. Total. Environ.* **586**, 1237-1244 (2017).
5. Liu, L. et al. The chemical-microbial release and transformation of arsenic induced by citric acid in paddy soil. *J. Hazard. Mater.* **421**, 126731 (2022).
6. Podgorski, J. & Berg, M. Global threat of arsenic in groundwater. *Science* **368**, 845-850 (2020).
7. Nagakubo, S., Nishino, K., Hirata, T. & Yamaguchi, A. The putative response regulator BaeR stimulates multidrug resistance of *Escherichia coli* via a novel multidrug exporter system, MdtABC. *J. Bacteriol.* **184**, 4161-4167 (2002).
8. Guo, T. et al. Increased occurrence of heavy metals, antibiotics and resistance genes in surface soil after long-term application of manure. *Sci. Total. Environ.* **635**, 995-1003 (2018).
9. Tang, X. et al. Effects of long-term manure applications on the occurrence of antibiotics and antibiotic resistance genes (ARGs) in paddy soils: Evidence from four field experiments in south of China. *Soil Biol. Biochem.* **90**, 179-187 (2015).
10. Zhang, N. et al. Coexistence between antibiotic resistance genes and metal resistance genes in manure-fertilized soils. *Geoderma* **382**, 114760 (2021).

Reviewer #2 (Remarks to the Author):

The authors' analysis of 511 global agricultural soil metagenomes shows that organic fertilization significantly increases the variety and quantity of groups carrying antibiotic and metal resistance genes compared to non-organic soils. This fertilization also affects the co-occurrence and expression of these genes, especially under arsenic stress, impacting soil properties and gene abundances. Moreover, through machine learning, the study investigated the distribution and potential pathogenic risks of these genes in agricultural lands, ultimately identifying high-risk areas in Central North America, Eastern Europe, Western Asia, and Northeast China.

This is a very interesting study. I really enjoy reading it. The figures are well presented, and the text is mostly written in a way that it is easy to follow for the reader (exceptions below). I have a few minor comments before I can recommend the paper for publication.

Response: Thank you very much for your comments. We sincerely appreciate your helpful suggestions. We have revised them one by one accordingly.

Minor comments:

1. The pipelines of each analysis are not provided.

Response: Thank you. All the pipelines and related codes have been provided online on GitHub (https://github.com/SiyuZhang1989/NC_Co-select_ARG-MRG).

2. Line 22: “Antibiotic (ARGs)” should be “Antibiotic Resistance Genes (ARGs)”.

Response: This has been corrected in line 23.

3. There is no reference to Fig. 2c in the manuscript.

Response: Thank you. We have cited Fig. 2c in lines 162, 164, and 166.

4. Fig. 5c: the “Relative abundance of MAGs (%)” should be “Relative abundance of MAGs (%)”.

Response: Thank you. We have corrected this in line 806.

REVIEWERS' COMMENTS

Reviewer #1 (Remarks to the Author):

All the comments and suggestions have been addressed. I am satisfied with the current version.

Reviewer #2 (Remarks to the Author):

I think authors have thoroughly revised the manuscript and addressed all my comments. The authors have provided clear evidence showing the influence of organic fertilization on the genetic-linkage of ARG-MRG. I believe this global map of the distribution of co-existent ARG-MRGs from agriculture lands will significantly extend the knowledge of the relevant research area. The job has been very well done, and I have no further questions for this revised version.